# Silver Nimesulide Complex in Bacterial Cellulose Membranes as an Innovative Therapeutic Method for Topical Treatment of Skin Squamous Cell Carcinoma

**DOI:** 10.3390/pharmaceutics14020462

**Published:** 2022-02-21

**Authors:** Tuany Zambroti Candido, Raphael Enoque Ferraz de Paiva, Mariana Cecchetto Figueiredo, Lilian de Oliveira Coser, Silmara Cristina Lazarini Frajácomo, Camilla Abbehausen, Izilda Aparecida Cardinalli, Wilton Rogerio Lustri, João Ernesto Carvalho, Ana Lucia Tasca Gois Ruiz, Pedro Paulo Corbi, Carmen Silvia Passos Lima

**Affiliations:** 1Department of Anesthesiology, Oncology and Radiology, Faculty of Medical Sciences, University of Campinas-UNICAMP, Campinas 13083-887, SP, Brazil; tuanyzcandido@gmail.com; 2Institute of Chemistry, University of Campinas-UNICAMP, Campinas 13083-970, SP, Brazil; raphael.enoque@gmail.com (R.E.F.d.P.); camilla@unicamp.br (C.A.); ppcorbi@iqm.unicamp.br (P.P.C.); 3Postgraduate Program in Medical Sciences, Faculty of Medical Sciences, University of Campinas-UNICAMP, Campinas 13083-887, SP, Brazil; maricecchettof@gmail.com; 4Department of Clinical Pathology, Faculty of Medical Sciences, University of Campinas-UNICAMP, Campinas 13083-887, SP, Brazil; coser.lilian@gmail.com; 5Department of Biological and Health Sciences, University of Araraquara-UNIARA, Araraquara 14801-320, SP, Brazil; scfrajacomo@uniara.edu.br (S.C.L.F.); wrlustri@uniara.edu.br (W.R.L.); 6Boldrini Children’s Center, Department of Pathology, Campinas 13083-210, SP, Brazil; izildaacardinali@gmail.com; 7Faculty of Pharmaceutical Sciences, University of Campinas-UNICAMP, Campinas 13083-871, SP, Brazil; carvalho@fcf.unicamp.br (J.E.C.); ana.ruiz@fcf.unicamp.br (A.L.T.G.R.)

**Keywords:** skin carcinoma, silver, nimesulide, bacterial cellulose, topical administration, antitumor activity

## Abstract

Oxidative stress and inflammation act on skin squamous cell carcinoma (SSCC) development and progression. Curative therapy for SSCC patients is mainly based on surgical resection, which can cause various sequelae. Silver ions have in vitro activities over tumor cells, while nimesulide has antioxidant and anti-inflammatory activities. This study aimed to evaluate the effects of a silver(I) complex with nimesulide (AgNMS) incorporated in a sustained release device based on bacterial cellulose membrane, named AgNMS@BCM, on topic SSCC treatment. The antiproliferative effect of AgNMS complex was evaluated in the SCC4, SCC15 and FaDu SCC lines. AgNMS complex activity on exposure of phosphatidylserine (PS) residues and multicaspase activation were evaluated on FaDu cells by flow cytometry. The AgNMS@BCM effects were evaluated in a SSCC model induced by 7,12-dimethylbenzanthracene/12-*o*-tetradecanoyl-phorbol-13-acetate (DMBA/TPA) in mice. Toxicity and tumor size were evaluated throughout the study. AgNMS complex showed antiproliferative activity in SCC15 and FaDu lines in low to moderate concentrations (67.3 µM and 107.3 µM, respectively), and induced multicaspase activation on FaDu cells. The AgNMS@BCM did not induce toxicity and reduced tumor size up to 100%. Thus, the application of AgNMS@BCM was effective and safe in SSCC treatment in mice, and can be seen as a potential and safe agent for topic treatment of SSCC in humans.

## 1. Introduction

Skin squamous cell carcinoma (SSCC) is the second most prevalent of all human cancers worldwide [1]. The tumor originates in keratinocytes of the basal extract of the epidermis and occurs in areas exposed to the sun, such as ears, face, scalp, and neck [2]. Solar ultraviolet (UV) radiation is the main cause of SSCC. UV radiation of the sun can directly modify the DNA structure of keratinocytes by formation of cyclobutanepyrimidine dimers and pyrimidine-pyrimidone photoproducts or indirectly by formation of reactive oxygen species (ROS) [3,4,5], leading to the emergence of SSCC as a consequence. UV radiation can also act on initiation, promotion, and progression of SSCC indirectly by the induction of inflammation, where the cyclooxygenase-2 (COX-2) protein has unequivocal importance [6,7,8,9,10,11,12].

The treatment of small SSCC is based on the resection of the tumor by surgery, but special attention is required for treatment of advanced tumors. Patients with advanced SSCC are often ineligible for either or both of curative surgery and radiotherapy, which may cause functional abnormalities, disfiguring and psychological problems [13,14,15] these patients are treated with cisplatin-based chemotherapy [14,16,17,18] or with the targeted EGFR inhibition cetuximab [14,19], which offers modest clinical benefits and potentially severe toxicity. Cemiplimab, an anti-PD-1 systemic agent, is emerging as an efficacious and tolerable systemic option for advanced SSCC, but its effects have been assessed only through phase I and II trials [14,20,21]. Thus, a topical treatment would be desirable to induce complete remission or reduction of tumor size, avoiding surgery or extensive surgery.

Silver has been used for treatment of human conditions since antiquity [22,23,24]. Although the use of this metal was purely empirical for a long time, nowadays the rational applications of this metal in medicine are closely related to its antibacterial [25,26] and anti-inflammatory activity [27]. Silver(I) complexes, particularly with *N*-heterocyclic carbenes (NHCs), have been shown to inhibit SCC cells in vitro [28,29,30,31]. Other proposed biomolecular targets for silver(I) complexes are nucleic acids and other proteins [32,33,34,35,36,37].

Nimesulide (NMS), a sulfonamide of the class of non-steroidal anti-inflammatory drugs (NSAIDS), has been commonly used in patients with inflammatory processes of respiratory tract and oral cavity, tendonitis and rheumatoid arthritis [38].

It inhibits COX-2 [39,40,41] and has antioxidant action, as can reduce the formation of reactive oxygen species (ROS) [42,43]. NSAIDS collectively have the potential to prevent the SSCC in humans [44,45]. NMS has in vitro antitumor activity against SCC [46,47], but its role in SSCC prevention and/or treatment is still unclear. 

In this context, our group developed a new silver complex with NMS (AgNMS) and started the evaluation of its multiple pharmacological activities [48]. This development expands on the important and classical class of “silver sulfonamides”, in analogy to the classical compound silver sulfadiazine [49] that combines a sulfonamide with silver to treat topically skin bacterial infections.

It is also important to mention that a growing interest in the topic treatment of SSCC has been observed in the last decade. The aim of this kind of treatment is to avoid mutilations imposed by surgery and toxic effects of systemic treatments [50,51,52,53,54]. The most common topic agents used in treatment of SSCC patients is 5-fluorouracil (5-FU) [55,56,57] and imiquimod [57]. 5-FU is a chemotherapeutic agent that blocks the thymidylate synthase and subsequently DNA production and cell proliferation. The main benefits of 5-FU in SSCC are the high response rate, few side effects, and its low cost when compared to other agents. However, it is not commercially available in all countries; thus, formulations in local stores/laboratories may not always guarantee high quality [55,56,57]. Imiquimod is an immune response modifier that activates Toll-like receptor 7 and stimulates interferon-α, interleukin-6, and tumor necrosis factor-α, generating inflammatory response and activation of different killer cells, such as macrophages, B-lymphocytes, and Langerhans cells. It seems to induce a good response in SSCC rate, but it may be associated with a higher risk of skin irritation and it is more expensive than other agents [57]. For this reason, the development of new devices for topical release of antitumoral drugs is also of great interest and bacterial cellulose membrane (BCM) emerges as an impressive biomaterial for this purpose [58,59].

Here, we described the evaluation of the in vitro antiproliferative activities of the AgNMS complex in thirteen different tumorigenic human cell lines, including SCC of tongue and pharynx, and in a non-tumoral line of immortalized human keratinocytes. We also evaluated if the topical administration of the AgNMS complex loaded in a transdermal device based on BCM, named AgMNS@BCM, in male Balb/c mice would bring local and/or systemic adverse effects as well as its efficacy in vivo studies in mice with induced SSCC. The promising results obtained so far can be seen in this manuscript for the first time.

## 2. Materials and Methods

### 2.1. Samples Preparation and Characterization

#### 2.1.1. Preparation of AgNMSComplex

The AgNMS complex was prepared according to a procedure previously reported by de Paiva et al. [48]. Briefly, to 10.0 mL of an aqueous solution containing 8.0 × 10^−4^ mol (0.2466 g) of NMS and 8.8 × 10^−4^ mol (0.0494 g) of KOH were added 2.0 mL of an aqueous solution containing 8.8 × 10^−4^ mol (0.1495 g) of AgNO_3_. The system was maintained under stirring for 1 h, at room temperature and protected from ambient light. The bright yellow solid obtained was isolated by vacuum filtration, washed with cold water and dried in a desiccator under vacuum. The full spectroscopic characterization of the compound can be found in the literature [48]. Elemental analysis results confirmed the composition of the AgNMS complex. Anal. Calcd. for AgC_13_H_11_N_2_O_5_S (%): C 37.6, H 2.7, N 6.7. Found (%): C 37.8, H 2.4, N 6.7.

#### 2.1.2. Preparation and Processing of BCM

For BCM production, *Gluconacetobacter hansenii* (ATCC 23769) was cultivated in FRU medium (fructose 60 g L^−1^, yeast extract 5.6 g L^−1^, ethanol 50 mL L^−1^, pH 4.5) and incubated in bio-oxygen demand at 28 °C for 72 h, as described by Lazarini et al. [58]. After that, an inoculum (1.0 McFarland in the nephelometric scale) was cultivated in culture flask containing 500 mL of molasses (ML) medium (sugarcane molasses 60 g L^−1^). The flask was statically incubated for 7 days at 28 °C in bio-oxygen demand incubator, with forced air convection produced horizontally, until BCM production. The resulting BCM was washed with NaOH solution (0.5 mol L^−1^, 2 h) followed by distilled water at 60 °C until pH neutralization. The processed BCM was transferred to a polypropylene container containing deionized water and sterilized in a vertical autoclave (25 min, at 121 °C, 15 lb/inch pressure), and frozen at −80 °C for further evaluations. The BCM composition was confirmed by elemental analysis. Anal. Calc. for (C_6_H_10_O_5_)_n_ (%): C 44.4, H 6.23. Found (%): C 43.5, H 6.56.

#### 2.1.3. Production of AgNMS@BCM

The production of the AgNMS@BCM transdermal device was performed as described by Cândido et al. [60]. The frozen BCM was lyophilized using a Benchtop Lyophilizer, model L108. Under sterile conditions, the lyophilized BCM was cut into squares of 1 cm^2^ in area and then swollen with AgNMS solution in aqueous dimethylsulfoxide (DMSO) at 10% for a final concentration of 1 mgcm^2^. The swollen membrane was heated at 65 °C in a ventilated oven (Overtechnical brand) for 3 h, to promote the membrane fibers retraction, affording the AgNMS@BCM device.

#### 2.1.4. Characterization of BCM and AgNMS@BCM

The BCM was characterized by infrared absorption spectroscopy (FTIR) and X-ray diffraction (XRD). FTIR measurements were performed using an Agilent Cary 630 spectrometer. The measurements were made in the ATR (attenuated total reflectance) mode, from 4000 to 600 cm^−1^ and 64 scans. X-ray diffraction (XRD) data was acquired in a XRD7000 Shimadzu diffractometer using CuKα radiation (λ = 1.5418 Å). The 2*θ* ranged from 5.0° to 60° with step scan 0.02° and speed scan 2°/min. The crystallinity index of BCM was determined according to the literature [61], using equation:(1)$$CrI=\frac{I_{002}-I_{100}}{I_{002}}\;\times \;100$$
where *CrI* corresponds the relative degree of crystallinity, *I*_002_ is the maximum intensity of the (002) plane diffraction and *I*_100_ is the intensity of the amorphous halo at 2*θ* = 18°. 

The microscopic characterization of the BCM and AgNMS@BCM was performed by scanning electron microscopy (SEM) using a JEOL JSM-6360 LV microscope. For SEM analysis, the BCM and the AgNMS@BCM were coated with evaporated carbon.

#### 2.1.5. Retention Capacity and Release of AgNMS@BCM

The test was performed as described by Lazarini et al. [58] using the diffusion method on Mueller-Hinton agar (MH, Merck) (CLSI, 2018). After inoculum preparation [5.0 mL sterile brain-heart infusion (BHI, Merck), incubation for 18 h at 35–37 °C], a suspension of *Staphylococcus aureus* (ATCC 25923) at 0.5 McFarland (~1.5 × 10^8^ CFU mL^−1^) was prepared in sterile BHI medium and uniformly spread using sterile cotton swab on sterile Petri plates containing MH agar. The evaluation of the retention capacity and sustained release of the AgNMS@BCM device was performed as described by Lazarini et al. [59]. The AgNMS@BCM device (1 cm^2^) was immediately placed on the surface of the solid agar with the *S. aureus* inoculum (time zero). The plate was incubated for 24 h at 35–37 °C. After 24 h, the inhibition zone was measured using caliper and then the discs were transferred to another MH plates inoculated with *S. aureus* (ATCC 25923). This procedure was repeated every 24 h until no detection of bacterial growth inhibition zones. The experiment was performed in triplicate. The maintenance of the inhibition zone after discs transferred every 24 h allowed determining the retention capacity and sustained release of the AgNMS@BCM. 

### 2.2. In Vitro Experiments

#### 2.2.1. Cell Lines Culture

Thirteen tumorigenic human cell lines [glioblastoma (U251), melanoma (UACC-62), mammary adenocarcinoma (MCF-7), multidrug-resistant high-grade ovarian serous adenocarcinoma (NCI-ADR/RES), renal cell carcinoma (786-0), large cell lung carcinoma (NCI-H460), adenocarcinoma of prostate (PC-3), high-grade ovarian serous adenocarcinoma (OVCAR-03), rectosigmoid adenocarcinoma (HT-29), chronic myelogenous leukemia (K562), squamous cell carcinoma (SCC) of the tongue (SCC4 and SCC15), SCC of the pharynx (FaDu)], and one non-tumorigenic human keratinocyte (HaCaT) cell lines were analyzed in the study. The SCC4, SCC15 and HaCaT were provided by Piracicaba Dental School of University of Campinas-UNICAMP (Piracicaba, SP, Brazil); the FaDu was provided by Odontology School of University of São Paulo-USP (São Paulo, Brazil) while the other human tumor cell lines were obtained from the National Cancer Institute 60 panel (NCI-60) and were provided by Frederick Cancer Research & Development Center, National Cancer Institute, Frederick, MA, USA. 

All cell lines were cultivated in complete medium [Roswell Park Memorial Institute (RPMI) 1640, Gibco^®^, Grand Island, NY, USA], 5% fetal bovine serum (code number 16000, Gibco^®^, Grand Island, NY, USA) and 1% penicillin:streptomycin mixture (Vitrocell^®^, Campinas, SP, Brazil; 1000 U·mL^−1^:1000 mg mL^−1^) at 37 °C, in 5% CO_2_. The experiments were done with cells at passage 4 to 10.

#### 2.2.2. Sample Preparation

The AgNMS complex and free NMS were diluted in DMSO (100 mg mL^−1^) followed by serial dilution in complete medium affording the final concentrations of 0.25, 2.5, 25, and 250 µg mL^−1^. Doxorubicin (0.025, 0.25, 2.5, and 25 µg mL^−1^) (Eurofarma, Itapevi, SP, Brazil) was used as positive control and was diluted in the same way. 

#### 2.2.3. Antiproliferative Activity Assay

A panel of ten tumorigenic human cell lines (U251, UACC-62, MCF-7, NCI-ADR/RES, 786-0, NCI-H460, PC-3, OVCAR-03, HT-29, K562) and one human keratinocytes line (HaCaT) were used for antiproliferative screening of the AgNMS complex and free NMS. After, a specific panel composed by SCC4, SCC15, and FaDu was used to verify the antiproliferative activity of the AgNMS complex against SCC. 

Cells in 96-well plates (100 μL/well, inoculation density: 3.5 to 6 × 10^4^ cell mL^−1^) were exposed to the four concentrations of each sample or doxorubicin (100 μL/well) in triplicate, for 48 h at 37 °C and 5% of CO_2_. Before (T0 plate) and after the sample addition (T1 plates), the cells were fixed with 50% trichloroacetic acid (50 μL/well), and the cell proliferation was determined by the spectrophotometric quantification (540 nm) of the cellular protein content using the sulforhodamine B assay (Sigma Aldrich, MO, USA). The analysis was performed in VersaMax spectrophotometer (Molecular Devices, San Jose, CA, USA). Considering the difference between absorbance values of untreated cells at 48 h (T1) and 0 h (T0, time of sample addition) as 100% of cell growth, the cell growth was determined for each cell line treated with each sample concentration. The results were plotted in a cell growth-concentration curve. Based on the curves, the concentration required to total growth inhibition (TGI) for each cell line was determined through sigmoidal regression using Origin 8.0 software (Origin Lab Corporation, Northampton, MA, USA) [62,63]. 

#### 2.2.4. Mechanism of Cell Death Induced by AgNMS

After culture in T75 flasks until 80% of confluence, the FaDu cells were treated with AgNMS solution (at 3.27 μg mL^−1^, in duplicate) for 24 and 48 h. Untreated cells (in duplicate) were used as negative control. After exposure, cells were harvested with 0.25% trypsin-EDTA and washed with phosphate-buffered saline solution (PBS). After centrifugation (10 min, 25 °C, 300× *g*), supernatants were discarded, and cell pellets were resuspended in PBS to prepare at least two aliquots with 1 × 10^6^ cell per 100 µL of each cell population (AgNMS-treated and untreated cells, in duplicate).

For the phosphatidylserine residues (PS) externalization assay, one aliquot of each cell suspension was treated with Guava Nexin^®^ reagent (mixture of Annexin V-PE with 7-AAD, 100 µL, catalog number 4500-0450) and kept in dark for 20 min. For multicaspases activation assay, the second aliquot of each cell suspension was treated with Guava Multicaspase reagent [100 µL, previously diluted 1:25 in phosphate buffer, catalog number FAM (carboxyfluorescein fluorochrome) 4500-0540] and incubated for 1 h at 37 °C. After that, the apoptosis wash buffer was added and the samples were centrifuged (25 °C, 10 min, 300 rpm). Then, each pellet was resuspended in 7-AAD reagent (200 µL, previously diluted 1:40 in phosphate buffer) and incubated for 10 min at room temperature. All Guava reagents were purchased from Millipore (Darmstadt, Germany). Marked cells were analyzed (50,000 events per replicate) in the FACSVerse cytometer (Becton Dickinson, CA, USA) and the FlowJo software was used in analysis.

### 2.3. In Vivo Evaluation

#### 2.3.1. Animals

Specific pathogen-free male Balb/c mice (*Mus musculus*, 6 weeks old, 68 animals) were obtained from the Multidisciplinary Center for Biological Research in the Field of Laboratory Animal Science, CEMIB/UNICAMP, after approval of the Animal Ethics Committee (CEUA/UNICAMP), protocol number 4141-1, (approval date: 11 April 2016). Animal care, as well as experimental protocols, were performed in accordance with the principles and guidelines adopted by the Brazilian National Council for Animal Experiment Control (CONCEA) and European Union Regarding Animal Experimentation (Directive of the European Counsel 86/609/EC). The mice were housed in a laboratory facility, in polyethylene boxes (matte white, 10 mice/cage) with disinfected softwood beddings, at room temperature of 22 ± 2 °C; relative humidity of 50 ± 20% and light/dark cycle of 12 h/12 h, during the adaptation period. All mice received pelleted feed Nuvilab^®^ and potable water ad libitum. The experiment begun when animals were 10-week-old weighing near 25 g. For experiments, the animals were randomly distributed two animals per cage in polyethylene boxes (matte white) divided in two compartments by a transparent acrylic wall that allowed visual and olfactory contact between mice (Appendix A).

#### 2.3.2. Repeated Dose 21-Day Topic Toxicity Study

Twenty-eight male Balb/c mice were trichotomized (9 cm^2^) under anesthesia (ketamine 120 mg kg^−1^ and xylazine 16 mg kg^−1^, intraperitoneal, i.p.) and randomly distributed into two experimental groups named G1 (BCM device, *n* = 14 animals) and G3 (AgNMS@BCM device, *n* = 14 animals). All animals were subjected to trichotomy every 15 days for 20 weeks and the body weights were weekly recorded. Photographic records at ≈20 cm of distance from animal, without using zoom, were made once a week.

At the 20th experimental week, the treatment with unloaded BCM (G1) and loaded AgNMS@BCM (G3) transdermal devices started. For better fixation of the membrane to dorsal area of each animal, a 3M micropore was used. The devices were changed every three days during 21 days of treatment. At the 24th experimental week, half of the animals of each group (*n* = 7, G1-day 21 and G3-day 21) were anesthetized (ketamine/xylazine 120:16 mg kg^−1^, i.p.) for blood samples collection (500 µL) from retroorbital plexus. After euthanasia by deepening anesthesia (ketamine/xylazine 300:30 mg kg^−1^, i.p.) followed by cervical dislocation, the back-skin portion, and some internal organs (liver, kidneys, testicles, and spleen) were removed, weighted, and conserved in buffered formalin solution. The remaining animals of each group (*n* = 7, G1-day 42 and G3-day 42) were clinically observed without receiving any additional treatment for 21 days, and then euthanized after deep anesthesia at the end of this period and submitted to the same evaluations of blood, skin, and organs collection.

#### 2.3.3. Skin Carcinogenesis and Treatment with AgNMS@BCM

Following a similar schedule, forty male Balb/c mice were trichotomized (9 cm^2^) under anesthesia (ketamine 120 mg kg^−1^ and xylazine 16 mg kg^−1^, i.p.) and randomly distributed into two experimental groups named G2 (BCM device, *n* = 20 animals) and G4 (AgNMS@BCM device, *n* = 20 animals). Twenty-four hours after trichotomy, animals of both groups were topically treated with 7,12-dimethylbenzanthracene (DMBA, single dose, 60 μg in 200 μL of acetone per animal). One week after the DMBA application, animals in G2 and G4 groups were topically treated with 12-*o*-tetradecanoyl-phorbol-13-acetate (TPA, 4 μg in 200 μL of acetone per animal), three times a week for 20 weeks [64]. All animals were subjected to trichotomy every 15 days and the body weights were weekly recorded during the experiment. The animals were clinically evaluated three times a week to verify the appearance of the skin tumors. 

The treatment with the unloaded BCM and the loaded AgNMS@BCM transdermal devices started at the 21th experimental week. The fixation of the AgNMS@BCM to dorsal area of animals and device changes were performed as described above. Once a week, photographic records at ≈20 cm of distance from animal, without using zoom, were made until 27th experimental week. For relative lesion area, the photos were analyzed using ImageJ software and considering a standard square rule for control area. At the 24th experimental week, half of the animals of each group (*n* = 10, G2-day 21 and G4-day 21) were anesthetized (ketamine/xylazine 120:16 mg kg^−1^, i.p.) and blood samples collection (500 µL) were obtained from retroorbital plexus. After euthanasia by deepening anesthesia (ketamine/xylazine 300:30 mg kg^−1^, i.p.) and cervical dislocation, the back-skin portion and liver, kidneys, testicles, and spleen were removed, weighted, and conserved in buffered formalin solution. The remaining animals of each group (*n* = 10, G2-day 42 and G4-day 42) were clinically observed for 21 additional days, without receiving any treatment. At the 27th experimental week, the remaining mice were euthanized and submitted to the same evaluations of blood, skin, and organs collection.

#### 2.3.4. Hematological Analysis

Blood samples (G1-G4) were collected in microtubes containing 10% potassium ethylenediaminetetraacetate (EDTA) solution (1 drop/tube) and analyzed in the automated hematology analyzer pocH-100iV^®^ (Sysmex, Lincolnshire, IL, USA), using program for mice and rats. The hematimetric parameters analyzed were complete count of red blood cells (RBC), white blood cell (WBC), and platelets (PLT) together with hemoglobin (HGB), hematocrit (HCT), mean corpuscular volume (MCV), mean corpuscular hemoglobin (MCH) determinations. 

#### 2.3.5. Histological Analysis

Fragments of the mice skin (G1–G4) were fixed with 4% formaldehyde for 24 h followed by 70% EtOH. After paraffin inclusion and slide preparation, the skin fragments were stained with hematoxylin/eosin and were observed in optic microscope Nikon Eclipse E200 Trinocular at 40×by an experienced pathologist. Representative microphotographs were acquired using a Bel Photonics IS1000 10.0 MP camera and the software Bel View Version 7.3.1.7. 

#### 2.3.6. Statistical Analysis

The results of flow cytometry analysis were analyzed by two-way ANOVA followed by Tukey’s test. The results of the in vivo evaluation were analyzed by one-way ANOVA followed by Bonferroni’s test. Significant differences were considered when *p*-value was <0.05. Statistical analyses were done using the Statistical Ultimate Academic software (TIBCO, Palo Alto, CA, USA) or GraphPadPrism software version 5.0 (GraphPad Software, San Diego, CA, USA).

## 3. Results

### 3.1. Characterization of BCM and AgNMS@BCM

The characterization of the BCM by FTIR spectra showed typical peaks [55] in the range 3350–3500 cm^−1^ corresponding to O–H stretching, 2800–2900 cm^−1^ corresponding to C–H stretching, 1160 corresponding to C–O–C stretching and 1035–1060 cm^−1^ corresponding to C–O stretching (Appendix A). The peak-by-peak comparison of the IR spectra of AgNMS@BCM with the free AgNMS and BCM showed that in the region < 2000 cm^−1^ all the peaks correspond to the unshifted vibrations observed for the precursor materials. The XDR (Appendix A) pattern showed the typical profile and crystallinity degree of BC [59]. The main diffraction peaks were found at 2*θ* 14.7, 16.7 and 22.7, and assigned to diffraction planes (101), (101) and (002) respectively. 

The pristine BCM (without the AgNMS complex) and the AgNMS@BCM (with the AgNMS complex) were well characterized by SEM (Figure 1A,B). More details on the microscopic morphology of the BCM and the AgNMS@BCM are given in Appendix A, respectively.

The transdermal device was able to maintain the AgNMS release up to 216 h uninterrupted, based on the disc diffusion assay developed in house. The released AgNMS led to the inhibition of *S. aureus* growth (12 mm halo throughout the experiment (Appendix A).

### 3.2. In Vitro Experiments

#### 3.2.1. Antiproliferative Activity of AgNMS

The AgNMS complex totally inhibited the growth of adenocarcinoma of breast (MCF-7, TGI = 26.3 µM), prostate (PC-3, TGI = 22.8 µM), ovary (OVCAR-03, TGI = 22.5 µM), and colon (HT-29, TGI = 41.1 µM) cell lines; for the other cell lines, the AgNMS complex showed anti-proliferative effect at concentrations higher than 70 µM. Free NMS was inactive for all human tumors (TGI > 200 µM). Both the AgNMS complex and NMS did not affect the proliferation of immortalized human keratinocytes HaCaT (Table 1, Appendix A).

The AgNMS complex selectively inhibited the SCC of tongue (SCC15, TGI = 67.3 µM) and melanoma (UACC-62, TGI = 2.8 µM) cells proliferation, being less active against SCC of pharynx (FaDu, TGI = 107.2 µM) and SCC of tongue (SCC4, TGI > 400) cell lines (Table 2; Appendix A).

##### Mechanism of Cell Death Induced by AgNMS

At the experimental conditions, the AgNMS complex did not induce significant PS exposure in FaDu cells. At 100 µM, the AgNMS complex induced significant increase in caspases activation without loss of membrane permeability (37.2%, 24 h-exposure; 25.5%, 48 h-exposure) in comparison to untreated FaDu cells (18.3%) (Appendix A; Figure 2).

### 3.3. In Vivo Experiments

#### 3.3.1. Toxicity Study

Considering body weight evolution, the DMBA/TPA challenge (groups G2 and G4) did not induce significant alteration in mice body weight in comparison to naïve mice (groups G1 and G3). At the 22nd experimental week, all groups showed a reduction in body weight gain that was recovered until 27th experimental week. Similar body weight variation was observed in all groups of mice (G1–G4) during the membrane application period (Appendix A; Figure 3).

The relative weight of liver, kidney, testicles, and spleen showed no significant changes induced either by the DMBA/TPA challenge (comparing mice in groups G1 and G2) neither by AgNMS@BCM treatment (comparing mice in groups G1, G3 and G4) at both two experimental endpoints (24th and 27th experimental week) (Table 3). It was observed only a slight significant reduction in total number of leukocytes, at the 24th experimental week, in AgNMS@BCM-treated mice without skin cancer induction (G3) in comparison to unloaded BCM-treated to mice without tumors (G1). This reduction was reverted by treatment discontinuity and was not observed in AgNMS@BCM-treated mice submitted to skin cancer induction (G4) (Table 3).

No macroscopic abnormalities in skin of mice from groups G1 and G3 were seen before, during and after treatment with unloaded BCM and AgNMS@BCM device both at 24th and 27th experimental week.

#### 3.3.2. Carcinogenesis and Effects of AgNMS@BCM in SCC

The chemically induced skin carcinogenesis resulted in 60% of animals (10 in 17 animals/group) with macroscopically detected lesions at 21st experimental week in G2 and G4 groups. Three animals per group of each group died during the experiments by reasons unrelated to experimental conditions. The lesions showed a wide range of sizes (from 0.5 to 20% of relative area at 21st experimental week), as presented in Table 4 and Figure 4A. Untreated animals (G2) showed an increase of 3.2% per week, in average, in the relative area of the lesions while AgNMS@BCM-treated animals presented an evolution rate of 0.9% per week, in average, representing 73% of reduction in the relative area evolution rate (Table 4, Figure 4B). Moreover, four animals in G4 with relative lesion area between 0.5 to 4% at the beginning of AgNMS@BCM treatment showed complete remission of skin lesions at 24th experimental week (Table 4, Figure 4C).

Histological analysis of mice from groups G1 and G3 showed characteristics of normal skin (Figure 4D). Loss of polarity of keratinocytes, cellular atypia, loss of nucleus-cytoplasm ratio, central nucleolus, infiltrate of basement membrane, ulcerations, and atypical mitosis were observed in skin of mice from G2, and these findings were in agreement with the induction of a verrucous cell carcinoma, the equivalent of SSCC of humans, in mice (Table 5, Figure 4E). The AgNMS@BCM treatment promoted complete regression of tumor cells in mice from G4 group (Figure 4F).

## 4. Discussion

This study reports, for the first time, the in vitro and in vivo antitumor effects of the AgNMS complex as a promising and safe alternative for topic treatment of SSCC. Among the metals already evaluated in the synthesis of metallodrugs, silver has anti-inflammatory activity [27] and silver-containing complexes have received notoriety as antitumor agents in the last decade, mainly due to the synergistic effects of silver when combined with different ligands [28,29,30,31,37]. In this work, NMS was chosen as a ligand due to its earlier reported antitumor effects against SCC [46,47]. Also, the antioxidant [42,43] and anti-inflammatory [39,40,41] effects of NMS contributed for its selection as a ligand for silver ions.

The AgNMS complex synthesized was evaluated in vitro against a panel of several cancer cell lines and, in special, for a panel focused on SCC cells. The obtained results demonstrated its anti-proliferative activity. Considering that the NMS ligand was inactive for all tumor cell lines, the AgNMS anti-proliferative activity could be attributed to the formed complex. Moreover, the AgNMS complex evidenced a significant selectivity for some of the considered human tumor cell lines, such as breast, prostate, ovarian, melanoma, and squamous cells, without affecting the proliferation of non-tumor keratinocytes (HaCat cell line).

Since the AgNMS complex showed an interesting activity against SCC cells in this study, the development of new therapies for SSCC based on the complex is of great interest. Considering that modulation of cell death mechanisms is one of the targets of chemotherapy, the influence of the complex on PS residues externalization, multicaspases activation and membrane permeability were evaluated in FaDu cells. It was observed that the AgNMS complex was able to activate multicaspases without PS externalization. Based on the recommendations of the Nomenclature Committee on Cell Death 2018 [65], some regulated cell death mechanisms, such as anoikis and apoptosis (intrinsic and extrinsic) involve caspases activation followed by PS exposure. At the experimental conditions, the obtained results might suggest that increasing the AgNMS exposition to FaDu cells could result in PS externalization as a consequence of caspase activation. However, even after 48 h-exposure, the AgNMS complex did not induce externalization of PS. It is already well known that to promote cell death is one of the pharmacological strategies to reduce/eliminate tumor cells. However, there is more than one way to induce regulated cell death (RCD) in eukaryotes. Each mechanism of RCD involves a dedicated molecular machinery that could be modulated by chemical substances [66,67]. Considering the hallmarks of cancer [68] and the results obtained in our study, the AgNMS complex seemed to affect cell proliferation by a mechanism different from classical apoptosis. Moreover, as NMS is a known anti-inflammatory agent, it is possible that the results obtained in chemical-induced skin carcinogenesis model might be a combination of modulation of cell proliferation and reduction of inflammatory stimuli in the microenvironment. Further studies will complement the present work elucidating the molecular mechanisms of action of AgNMS as anticancer agent.

The promising in vitro anti-proliferative results obtained for the AgNMS complex against SCC and melanoma cells in this study and the previous experience of our group in controlled releasing devices based on BCM [58,59], prompted to the development of the AgNMS@BCM device. Considering the characteristics of the AgNMS complex and BCM, the total load capacity of membrane (1 mg cm^−2^) was chosen for this study. The release assay based on antimicrobial activity of the AgNMS complex previously described [48] demonstrated that the AgNMS@BCM device was able to keep a release of the complex up to 216 h. 

Aiming to evaluate the local and systemic toxicity and antitumoral effects, the AgNMS@BCM device was topically applied in mice, without and with DMBA/TPA-induced skin carcinogenesis. Local toxicity was not seen in this study. Considering body weight evolution during all the experiment, nonsignificant change (≥20%) [69] could be related to skin cancer induction or treatment with the pristine BCM and the AgNMS@BCM. Reduction on body weight gain between the 21st and 22nd experimental week was seen in all animals, and this finding could be attributed to the discomfort during the membranes applying. In fact, it is already well known that transient oscillations in the husbandry conditions can affect mice feeding and consequently body weight gain [70].

Macroscopic analysis (appearance and relative organ weight) of internal organs is an indicator of xenobiotic-induced microscopic alterations [71]. At the two experimental endpoints (24th and 27th experimental weeks), AgNMS@BCM-treated mice (G3 and G4) presented nonsignificant changes in relative liver, kidneys, spleen and testis weight in comparison to BCM-treated mice (G1). The DMBA/TPA application on mice skin (G2) did not induce any systemic alteration in comparison to G1 mice. Furthermore, either AgNMS@BCM treatment neither DMBA/TPA challenge induced significant alterations on hematimetric parameters in comparison to the pristine BCM-treated mice (G1) and the literature data for normal limits for the species [72,73]. These results might suggest that the AgNMS topically administrated did not reach systemic circulation at a toxic concentration. However, comparing animals without DMBA/TPA challenge (G1 and G3 groups), AgNMS@BCM-treated mice showed a slight reduction in total leukocyte count in comparison to unloaded BCM-treated animals. After discontinuation of treatment, normalization of the total leukocyte count (WBC) was observed, suggesting that the AgNMS complex was absorbed through whole skin. The reduction in WBC could be attributed to the anti-proliferative effect of the AgNMS complex on bone marrow cells, a common effect of several chemotherapeutic agents [74]. Despite the reduction, the WBC value for animals in group G3 was classified as normal in comparison to the normality range described in literature [72,73]. Interestingly, DMBA/TPA challenged mice treated with AgNMS@BCM device did not show any alteration in WBC in comparison to animals from G1 group.

The DMBA/TPA-induced skin carcinogenesis is a classical model characterized by the induction of benign squamous papillomas that can progress to verrucous cell carcinoma, the equivalent of SSCC of humans, in mice [75]. Together with the proliferative effect, TPA induces a tissue inflammatory reaction that potentialize the TPA-stimuli to cell proliferation [75]. These characteristics made DMBA/TPA-induced tumor model more like the human disease. In this study, around 60% of animals showed macroscopic tumors, and all of them presented histological characteristics of verrucous cell carcinoma. 

Interestingly, the untreated tumor presented an increasing rate in relative area of 4.4% per week while the AgNMS treatment promoted a reduction of almost 75% in this parameter. Moreover, four in ten animals presented total remission of lesions after the AgNMS@BCM treatment while in a single animal, the AgNMS@BCM device seemed not affect tumor progress. These differences should be explained by the heterogeneity of the DMBA/TPA-induced tumor model [75].

In fact, the effects of the AgNMS complex against SSCC were not a surprise. It is already well known that ROS production by solar UV radiation [3,4,5] and inflammation with participation of the cyclooxygenase-2 (COX-2) protein [6,7,8,9,10,11,12] are important metabolic processes involved in SSCC development and progression. Silver has anti-inflammatory activity [27] and silver complexes, particularly with N-heterocyclic carbenes (NHCs), inhibits the proliferation of the A431 (SCC) cells in vitro [28,29,30,31]. On the other hand, NMS reduces ROS formation [42,43], inhibits COX-2 protein [39,40,41], and has in vitro antitumor activity against SCC [46,47]. Thus, silver and NMS together in a complex may have acted collectively as an antitumoral agent in the DMBA/TPA-induced tumor model.

The effects of the AgNMS complex in SSCC treatment in a transdermal device was also expected in this study, since topic treatment of the SSCC with niosomal system, made up of alpha,omega-hexadecyl-bis-(1-aza-18-crown-6), for delivery of 5-fluorouracil [50], 5-fluorouracil, imiquimod, diclofenac, ingenolmebutate, retinoids, resiquimod, piroxicam, dobesilate, and betulinic acid [51] were early described. Also, drug-loaded nanostructured lipid carrier gel of quercetin and resveratrol [52], dermal targeted combinatorial lipid nanocolloidal based formulation of 5-fluorouracil and resveratrol [53], and silver nanoparticles with sericin and chitosan [54] have been noticed with great interest in the medical literature. It is worth mentioning that the low cost and complexity involved in the AgNMS@BCM preparation may confer an advantage for this device among others in the SSCC treatment.

The topic administration of 5-FU and imiquimod in the treatment of SSCC patients has been seen with enthusiasm, because high response rate and few side effects have been attributed to both agents [55,56,57], but comparisons of the magnitude of the effects of topic administration of AgNMS@BCM, 5-FU and imiquimod will only be possible after conducting studies with use AgNMS@BCM in humans.

## 5. Conclusions

This work represents the first investigation focusing the in vitro anti-proliferative activity and the in vivo local and systemic evaluation of the AgNMS complex as a promising anticancer agent. The AgNMS@BCM device was able to release the AgNMS complex in a controlled way circumventing the systemic toxicity. Taking all together, the AgNMS@BCM device has shown to have a high potential for SSCC treatment.Further studies are required to better understand of the mechanism of action of the complex as well as to know its effects in humans.

## Figures and Tables

**Figure 1 pharmaceutics-14-00462-f001:**
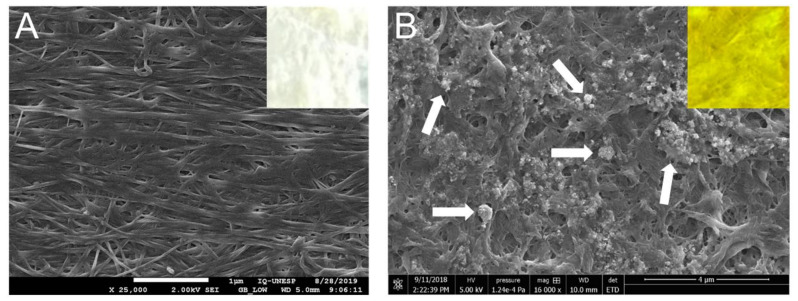
Scanning electron microscopy (SEM) show (**A**) the pristine BCM (at 25,000×) and (**B**) the AgNMS@BCM (16,000×). The white arrows in (**B**) indicate the AgNMS fragments impregnated into the networks of membrane fibers. The insets in (**A**,**B**) demonstrate that upon impregnation with the AgNMS, the bacterial cellulose membrane (originally white, inset (**A**)) acquires the characteristic yellow color of the AgNMS compound (inset (**B**)).

**Figure 2 pharmaceutics-14-00462-f002:**
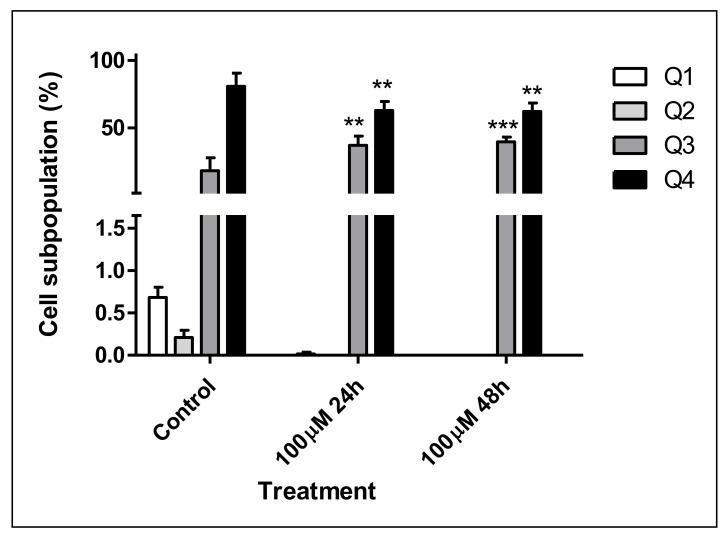
AgNMS-induced multicaspase activation on FaDu cells after 24 and 48 h exposure. Results expressed as mean (%) ± standard deviation from duplicates of 1 independent experiment. Treatment: control = untreated cells; AgNMS = silver nimesulide complex at 100 µM, after 24 and 48 h exposure. Cell subpopulation (%) for multicaspase activation: Q1 = FAM (−)/7-AAD (−); Q2 = FAM (+)/7-AAD (−); Q3 = FAM (+)/7-AAD (+); Q4 = FAM (−)/7-AAD (+); FAM: carboxyfluoresceinfluorochrome. Statistical analysis: two-way ANOVA followed by Tukey’s test (** *p* < 0.01, *** *p* < 0.001 comparing to control group).

**Figure 3 pharmaceutics-14-00462-f003:**
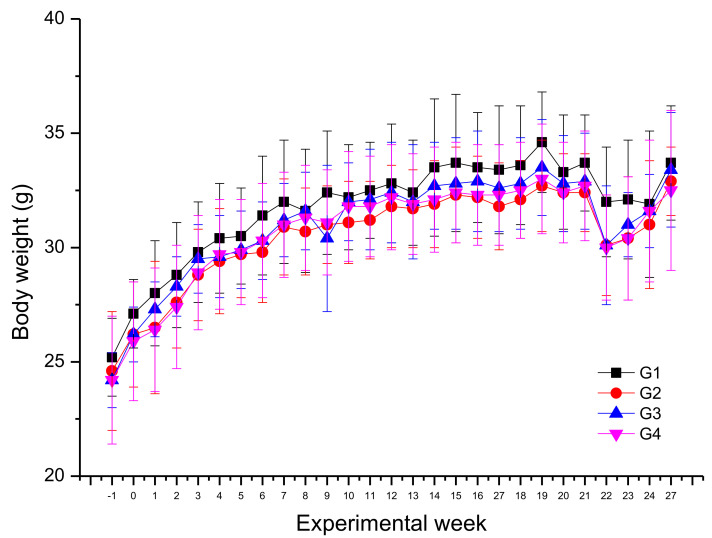
Body weight evolution of mice during all experimental weeks. Results expressed as mean (g) ± standard deviation (*n* = 20 animals/group, G2 and G4; *n* = 14 animals/group, G1 and G3) of mice (male Balb/c) body weight. Experimental weeks: S1 = 7,12-dimethylbenzanthracene (DMBA) treatment; S0–S20 = 12-*o*-tetradecanoyl-phorbol-13-acetate (TPA) treatment; S21–S23 = bacterial cellulose membrane (BCM) device treatment; S24–S27 = recovering time. Experimental groups: G1 = unloaded BCM device without skin cancer induction; G2 = unloaded BCM device with skin cancer induction; G3 = BCM with AgNMS complex (AgNMS@BCM) device without skin cancer induction; G4 = AgNMS@BCM device with skin cancer induction.

**Figure 4 pharmaceutics-14-00462-f004:**
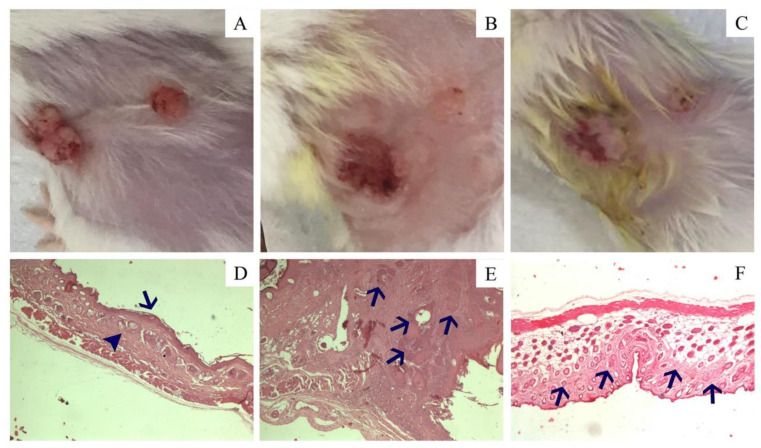
Representative photographs and microphotographs of male Balb/c mice treated with AgNMS complex in a topical adhesive based on bacterial cellulose membrane (AgNMS@BCM). Macroscopic findings: (**A**) skin with tumor before starting treatment with AgNMS@BCM, (**B**) skin with reduction of the tumor size after 21 days of treatment (day 21), and (**C**) skin 21 days after the end of treatment with AgNMS@BCM in a topical adhesive (day 42) showing complete remission of the tumor. Microscopic findings: (**D**) normal skin with epidermis (arrow) and dermis (arrowhead) without changes (HE, 400×), (**E**) verrucous cell carcinoma before treatment with clusters of tumor cells invading the dermis (arrows) (HE, 400×), and (**F**) skin after treatment with metal complex with areas of collagen fibrosis in the dermis (arrows) with regression of tumor cells (HE, 100×).

**Table 1 pharmaceutics-14-00462-t001:** Antiproliferative activity of doxorubicin, AgNMS complex and free NMS required to elicit total cell growth inhibition.

Cell Line	TGI (µM ± SD)		
	Doxo	AgNMS	NMS
U251	>46	102.1 ± 51.9	>800
MCF-7	11.8 ± 9.3	26.3 ± 8.6	>800
NCI-ADR/RES	>46	157.3 ± 75.1	230.7 ± 129.2
786-0	3.4 ± 0.2	79.8 ± 22.8	>800
NCI-H460	>46	116 ± 50	>800
PC-3	2.6 ± 1.6	22.8 ± 5.7	>800
OVCAR-03	<0.046	22.5 ± 24.5	269.6 ± 66.8
T-29	<0.046	41.1 ± 8.2	234.2 ± 68.8
K562	9.4 ± 10.9	>600	>800
HaCaT	<0.046	>600	>800

Results (two independent experiments) expressed as concentration followed by standard deviation (SD) required to elicit total growth inhibition (TGI), calculated by sigmoidal regression using Origin 8.0 software. Samples: Doxo = doxorubicin (positive control, concentration range: 0.025–25 µg mL^−1^); AgNMS = silver nimesulide complex (concentration range: 0.25–250 µg mL^−1^); NMS = nimesulide (concentration range: 0.25–250 µg mL^−1^). Human tumor cell lines: U251 (glioblastoma); MCF-7 (breast, adenocarcinoma); NCI-ADR/RES (multidrug-resistant high-grade ovarian serous adenocarcinoma); 786-0 (kidney, adenocarcinoma); NCI-H460 (lung, non-small cell carcinoma); PC-3 (prostate, adenocarcinoma); OVCAR-03 (high-grade ovarian serous adenocarcinoma); HT29 (rectosigmoid adenocarcinoma); K562 (chronic myelogenous leukemia). Human non tumor cell line: HaCaT (immortalized keratinocyte).

**Table 2 pharmaceutics-14-00462-t002:** Influence of doxorubicin and AgNMS complex on cell growth of squamous carcinoma and melanoma cell lines.

Cell Line	TGI (µM ± SD)	
	Doxo	AgNMS
SCC15	1.8 ± 0.6	67.3 ± 55.1
SCC4	3.5 ± 1.2	>400
FaDu	5.4 ± 2.3	107.2 ± 64.2
UACC-62	<0.046	2.8 ± 1.1

Results expressed as concentration followed by standard deviation (SD) required to elicit total growth inhibition (TGI), calculated by sigmoidal regression using Origin 8.0 software (one experiment in triplicate). Samples: Doxo = doxorubicin (positive control, concentration range: 0.025–25 µg mL^−1^); AgNMS = silver nimesulide complex (concentration range: 0.25–250 µg mL^−1^). Human tumor cell lines: SCC15 (squamous cell carcinoma of the tongue); SCC4 (squamous cell carcinoma of the tongue); FaDu (squamous cell carcinoma of the pharynx); UACC-62 (melanoma).

**Table 3 pharmaceutics-14-00462-t003:** Relative organ weight and hematological parameters of Balb/c mice in toxicological (G1 and G3) and chemically-induced skin carcinogenesis (G2 and G4) experiments at two endpoints (24th and 27th experimental week).

Group		G1		G2		G3		G4	
Experimental Day		21-Day	42-Day	21-Day	42-Day	21-Day	42-Day	21-Day	42-Day
Relative organ weight	Liver	5.0 ± 0.4	4.8 ± 0.2	5.6 ± 0.4	4.9 ± 0.5	4.9 ± 0.4	4.6 ± 0.2	5.4 ± 0.5	5.2 ± 0.3
	Kidneys	1.9 ± 0.1	1.9 ± 0.1	2.0 ± 0.1	1.87 ± 0.08	1.9 ± 0.1	1.9 ± 0.1	1.9 ± 0.2	1.9 ± 0.1
	Testis	0.72 ± 0.07	0.6 ± 0.2	0.7 ± 0.2	0.69 ± 0.07	0.74 ± 0.08	0.72 ± 0.02	0.7 ± 0.1	0.7 ± 0.1
	Spleen	0.47 ± 0.04	0.36 ± 0.09	0.6 ± 0.4	0.4 ± 0.1	0.45 ± 0.07	0.38 ± 0.04	0.6 ± 0.2	0.5 ± 0.2
Hematological parameters	WBC (×10^3^ μL)	6.1 ± 0.4	6.6 ± 1.6	6.5 ± 2.4	7.8 ± 0.9	3.8 ± 0.5 *	7.9 ± 2.7	5.5 ± 1.1	8.0 ± 1.4
	RBC (×10^6^ μL)	11.0 ± 0.2	11.1 ± 0.3	10.9 ± 0.6	11.0 ± 0.6	11.2 ± 0.4	11.3 ± 0.5	10.7 ± 0.8	10.7 ± 1.1
	HGB (g/dL)	15.0 ± 0.4	15.2 ± 0.5	14.8 ± 0.7	15.4 ± 0.7	15.2 ± 0.7	15.2 ± 0.8	14.5 ± 0.8	15. ± 0.8
	HCT (%)	53.4 ± 1.1	54.3 ± 1.7	53.3 ± 2.3	54.5 ± 2.7	54.6 ± 1.6	56.1 ± 2.7	52.3 ± 2.6	53.3 ± 5.7
	MCV (fL)	48.8 ± 1.2	48.9 ± 0.6	48.6 ± 1.1	49.6 ± 0.2	48.9 ± 0.6	49.6 ± 0.3	49.1 ± 1.5	49.7 ± 1.1
	MCH (pg)	13.7 ± 0.2	13.7 ± 0.4	13.3 ± 0.9	13.7 ± 0.2	13.6 ± 0.2	13.4 ± 0.1	13.6 ± 0.3	13.4 ± 0.6
	MCHC (g/dL)	28.2 ± 1.0	27.9 ± 0.6	27.3 ± 1.7	27.6 ± 0.4	27.9 ± 0.6	27.2 ± 0.1	27.7 ± 0.6	27.0 ± 0.7
	PLT (×10^3^μL)	1686 ± 92	1726 ± 155	1620 ± 304	1662 ± 178	1842 ± 272	1581 ± 27	1750 ± 308	1501 ± 126

Results expressed as mean ± standard deviation (*n* = 7 at each endpoint, G1 and G3; *n* = 10 at each endpoint, G2 and G4). Animal: male Balb/c mice (8-week-old at the beginning of the experiment). Statistical analysis: ANOVA followed by Tukey’s test (* *p* < 0.05). Challenge: DMBA (7,12-dimethylbenzanthracene, single dose at S-1, 60 μg in 200 μL of acetone per animal) followed by TPA (12-*o*-tetradecanoyl-phorbol-13-acetate, 4 μg in 200 μL of acetone per animal), three times a week for 20 weeks starting at S0 experimental week and trichotomy every 15 days = G2 and G4; trichotomy every 15 days without induction of carcinogenesis = G1 and G3. Treatments: unloaded bacterial cellulose (G1 and G2); AgNMS (silver nimesulide complex) at 1 mg/1 cm^2^ in bacterial cellulose membrane (AgNMS@BCM) (G3 and G4); Parameters: Relative organ weight = ratio between organ weight (liver, kidney, testis or spleen) and the respective mice body weight; Hematological parameters: WBC = white blood cell; RCB = red blood cell; HGB = hemoglobin; HCT = hematocrit; MCV = mean corpuscular volume; MCH = mean corpuscular hemoglobin; MCHC = mean cellular hemoglobin concentration; PLT = platelets.

**Table 4 pharmaceutics-14-00462-t004:** Relative lesion area evolution during the DMBA/TPA-induced skin carcinogenesis model in mice.

Group	Parameter	Experimental Week					
		21th	22th	23th	24th	27th	Δ Average ^c^
G2	RLA ^a^	11.4 ± 6.7	13.9 ± 8.1	13.1 ± 8.3	15.9 ± 8.4	19.3 ± 2.8	-
	Δ ^b^	-	2.5 ± 5.3	1.7 ± 4.8	4.5 ± 12.6	8.1 ± 9.4	3.2 ± 8.1
G4	RLA ^a^	3.3 ± 3.2 ***	3.0 ± 2.3 ***	5.0 ± 4.8 ***	3.3 ± 1.7 ***	5.2 ± 1.0 ***	-
	Δ^b^	-	0.0 ± 2.0	1.8 ± 3.4	−0.1 ± 2.1	2.5 ± 1.6	0.9 ± 2.7

Results expressed as mean ± standard deviation (*n* = 10 animals/group with macroscopic lesions from 17 animals) of relative lesion area in mice (male Balb/c). Parameters: ^a^ RLA = Relative lesion area, evaluated by comparing the lesion area to a standard square ruler photographed together with each mouse. Photos analyzed by using ImageJ software; ^b^ Δ = difference between RLA at an experimental week and at 21st experimental week (starting treatment); ^c^ Δ average = average increasing rate per week. Experimental weeks: S21–S23 = BCM device treatment; S24–S27 = recovering time. Experimental groups: G2 = unloaded BCM device with skin cancer induction; G4 = AgNMS@BCM device with skin cancer induction. Samples: BCM = unloaded bacterial cellulose membrane; AgNMS = nimesulide-silver complex; AgNMS@BCM = bacterial cellulose membrane loaded with AgNMS 1 mg/cm^2^. Statistical analysis: one-way ANOVA followed by Bonferroni’s test (*** *p* < 0.001).

**Table 5 pharmaceutics-14-00462-t005:** Histological characteristics of the normal skin epithelium and skin epithelium after induction of verrucous cell carcinoma.

Normal Skin	Skin with Induced Verrucous Cell Carcinoma
Preserved architecture	Architecture altered
Epidermis normal keratinized stratified squamous epithelium. Polarization of the layers of the preserved epidermis. No thickening	Altered epidermis, loss of polarity of keratinocytes, cellular atypia, loss of nucleus-cytoplasm ratio, evident central nucleolus
Intact basement membrane	Infiltrated basement membrane
No ulceration	present ulceration
Present and typical mitosis (bipolar)	Tripolar (Y) and tetrapolar (X) increased and atypical mitosis

## Data Availability

The dataset analyzed during the current study is available from the corresponding author Carmen S. P. Lima on reasonable request.

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
