# Peer review of "Silver Nimesulide Complex in Bacterial Cellulose Membranes as an Innovative Therapeutic Method for Topical Treatment of Skin Squamous Cell Carcinoma"

_pharmaceutics, 2022, doi:10.3390/pharmaceutics14020462_

Round 1

Reviewer 1 Report

Comments for the manuscript titled ‘Silver nimesulide complex in bacterial cellulose membranes as 2 an innovative therapeutic method for topical treatment of skin 3 squamous cell carcinoma’.

Authors have proposed the application of silver nimesulide, analogical model of sulfonamide combined with bacterial cell membrane as drug releasing agent to treat skin squamous cell carcinoma. The publication needs major revision before being considered for publication:

  1. Although sulfonamide have been mostly used as an antimicrobial and antifungal agent; topical agents like 5 -flurouracil (FU) and combinatorial approaches involving FU have been currently used as recommended topical therapy. Authors are suggested to create comparison with current topical treatment to reach a more informed conclusion on the better skin 3 squamous cell carcinoma (SCCC) treatment approach.
  2. In the context of Figure 1, authors are requested to provide better quality picture, depicting detailed macroscopic aspects of pristine BCM and AgNMS@BCM.
  3. Authors are suggested to provide a data for the porosity profile of BCM (loaded and unloaded), and before and after application (i. e.) in in vitro conditions and in-vivo
  4. Authors are requested to perform an western bot analysis of the tumor proteins (p53, p27, anti-bcl2caspase-3anti β actin) with NMS and AgNMS complex from the tissues.

Reviewer 2 Report

A very interesting and innovative original study showing for the first time the in vitro and in vivo antitumor effects of a silver and nimesulide complex
as a promising and safe alternative for the topic treatment of SSCC. Given the novelty of the topic, in a field where no drug seem to have an effect on this tumor, I think that this paper should be granted publication after minor revisions:

Line 62 you should add: "currently, no topical drug seem to have a clinically demonstrated effect in the management of SCC" and cite an article such as: doi: 10.3390/curroncol28040213. and doi: 10.3390/medicina57060563.

In the statistical analysis, didn't the Bonferroni correction lower the p-value to consider significant? Please check.

Round 2

Reviewer 1 Report

Regarding the requirement of protein expression analysis, authors are can show either ELISA or western blot, as these data would be evident about the functionality aspect, and revealing potential mechanism's validation.

Author Response

We inform the reviewer that, unfortunately, we will not be able to perform ELISA or western blotting of tumor proteins (p53, p27, anti-bcl2, caspase-3, and anti β actin) with and without the AgNMS complex at this time, as we do not have all the reagents in the laboratory and we will have difficulty obtaining them quickly during the coronavirus 19 pandemic. We have tried to obtain the reagents from other researchers and they all face the same problems in obtaining reagents during the pandemic. Thus, we apologize to the reviewer, but even knowing the importance of protein levels to show the potential mechanism of action of AgNMS, we were unable to perform these assays for this study. In addition, considering your request/suggestion, we considered it appropriate to improve the discussion about the possible effects of the AgNMS on squamous cell carcinoma and we added two pertinent references. Please, see the discussion section (page 14; lines 521 to 531, page 15, lines 603 to 604, page 16, lines 605 to 607) and reference section (page 20, references 66 and 67) of the current version of the manuscript. We appreciate the reviewer’s suggestion and inform that these experiments will certainly be performed in the next study conducted with this compound by our group.
